# NEURAL CIRCUIT ARCHITECTURAL PRIORS FOR EMBODIED CONTROL

## ABSTRACT

Artificial neural networks for simulated motor control and robotics often adopt generic architectures like fully connected MLPs. While general, these *tabula rasa* architectures rely on large amounts of experience to learn, are not easily transferable to new bodies, and have internal dynamics that are difficult to interpret. In nature, animals are born with highly structured connectivity in their nervous systems shaped by evolution; this innate circuitry acts synergistically with learning mechanisms to provide inductive biases that enable most animals to function well soon after birth and improve abilities efficiently. Convolutional networks inspired by visual circuitry have successfully encoded biases useful for vision tasks. However, it is unknown the extent to which ANN architectures inspired by neural circuitry can yield useful inductive biases for other domains. In this work, we ask what advantages biologically inspired network architecture can provide in the context of motor control. Specifically, we translate *C. elegans* circuits for locomotion into an ANN model applied to a simulated Swimmer agent. On a locomotion task, our architecture achieves good initial performance and asymptotic performance comparable with MLPs, while dramatically improving data efficiency and requiring orders of magnitude fewer parameters. Our architecture is also more interpretable and transfers to new body designs. An ablation analysis shows that principled excitation/inhibition is crucial for learning, while weight initialization contributes to good initial performance. Our work demonstrates several advantages of ANN architectures inspired by systems neuroscience and suggests a path towards modeling more complex animals.

## 1 INTRODUCTION

Artificial neural networks for simulated motor control and robotics often adopt generic architectures like fully connected multi-layered perceptrons (MLPs) (Pierson & Gashler, 2017; Levine et al., 2016; Bin Peng et al., 2020; Heess et al., 2016). While general, these *tabula rasa* architectures rely on large amounts of experience to learn. Data efficiency is especially important in motor control because, unlike computer vision and natural language processing which have greatly benefited from the availability of large datasets (Bommasani et al., 2021), gathering large-scale data for robotics is challenging. Experience must be gathered through interaction with the environment, which is difficult for reasons including time, human labor, safety, maintenance, and reproducibility (Kroemer et al., 2020). Transfer is also a significant problem, as experience is often specific to the robot body it was gathered on, and ANNs trained to control one body are not easily adapted a different one. In addition, *tabula rasa* architectures like MLPs are in general difficult to interpret, as they have internal dynamics that are distributed across units and non-trivial to relate to agent behavior (Merel et al., 2019a).

In nature, animals are born with highly structured connectivity in their brains and nervous systems that has been shaped over millennia by evolution (Zador, 2019). In some cases, this innate circuitry confers abilities with little or no learning; in others, it guides the learning process by providing strong inductive biases (Lake et al., 2017). These innate and learned mechanisms act synergistically, enabling most animals to function well soon after birth, while continuing to improve and acquire skills in a data-efficient manner, e.g. a horse learning to walk with only a couple hours of experience. Moreover, despite species-specific variations, there is a significant amount of shared architecture (e.g. cerebellum, basal ganglia) and design principles (e.g. hierarchical modularity, partial autonomy) even between distantly related species (Merel et al., 2019b). Biological circuit architecture is often highly

structured and sparse (Luo, 2021), in sharp contrast to the all-to-all connectivity of MLPs. Moreover, evolution has progressively built more advanced abilities on lower-level circuits, leveraging modular structure to transfer existing working designs to new animal bodies (Cisek, 2019; Merel et al., 2019b). Taken together, these findings from neuroscience suggest that structured neural circuits in animals instantiate efficient, transferable, and modular solutions for high-dimensional embodied control.

Capturing some of this structure in model architecture may enable ANNs to narrow the gap between artificial and natural systems. For example, convolutional networks inspired by visual circuitry have successfully encoded the inductive biases of spatial locality and weight sharing to improve performance, data efficiency, and parameter efficiency for vision tasks (Lindsay, 2021). Since their neuroscience-inspired origins, convolutional networks have benefited from novel layer types, activation functions, and modules (Gu et al., 2017), as well as novel architectures, e.g. LeNet (LeCun et al., 1989), AlexNet (Krizhevsky et al., 2012), VGG (Simonyan & Zisserman, 2015), ResNet (He et al., 2015). The advantages of (non-biologically inspired) architectural priors has also been established across other machine learning domains. In natural language processing, architectural priors have specialized to handle sequences with designs including recurrent networks, e.g. LSTM (Hochreiter & Schmidhuber, 1997), and attention networks, e.g. Transformer (Vaswani et al., 2017). However, it is unknown the extent to which ANN architectures inspired by neural circuitry can yield useful inductive biases for other domains.

In this work, we ask what advantages biologically inspired network architecture can provide in the context of motor control. Specifically, we translate *C. elegans* circuits for locomotion into an ANN model applied to a simulated Swimmer agent. Our architecture is an instance of what we call a "Neural Circuit Architectural Prior" (NCAP), to denote an ANN architecture that encodes prior structure / inductive biases inspired by biological neural circuits. In contrast to previous work on neuromechanical models of movement and central pattern generators (Sarma et al., 2018; Izquierdo & Beer, 2015; Jiao et al., 2021), our model is designed within the discrete-time ANN framework that is standard in machine learning and is fully differentiable, enabling us to train parameters with reinforcement learning (RL) and evolution strategies (ES), and directly investigate the role of network architecture by comparing to MLPs. Further, our model controls an agent body from a standard benchmark (i.e. not custom-designed to work with our architecture); this body is significantly different from *C. elegans* in terms of mechanics, degrees of freedom, and actuators.

On a locomotion task, our architecture achieves good initial performance and asymptotic performance comparable with MLPs, while dramatically improving data efficiency and requiring orders of magnitude fewer parameters. Our architecture is more interpretable and transferable to new body designs. An ablation analysis shows that principled excitation/inhibition significantly contributes to learning. Our work demonstrates several advantages of ANN architectures inspired by systems neuroscience and suggests a path towards modeling more complex animals.

The primary contributions of this work are:

1. A network architecture inspired by *C. elegans* circuits for locomotion, which combines the discrete-time ANN framework that is standard in machine learning with features from computational neuroscience like constraints on synapse sign (i.e. excitation vs. inhibition) and special cell types (i.e. intrinsic oscillators);

2. An evaluation of our model's initial performance, asymptotic performance, data efficiency, parameter efficiency, interpretability, and transfer compared to standard MLP architectures; and

3. An ablation analysis of the effects of sign constraints, weight initialization, weight sharing, and sparse connectivity on performance and learning.

Code and videos are available in the Supplementary Materials.

## 2 RELATED WORK

A robotic controller ultimately outputs generalized torques $\tau$ to apply at each actuator. A neural network controller can directly output torques (Levine et al., 2016), or it can output task-space positions $x$ or accelerations $\ddot{x}$ that are converted into torques through an analytic controller like operational space control (Khatib, 1987). However, some further level of abstraction and prior knowledge is often used to improve performance and simplify learning (Kroemer et al., 2020).

**Trajectory Priors**   Desired movement is encoded through equations of motion. Dynamic Movement Primitives (DMP) (Schaal, 2006; Pastor et al., 2009) use a set of differential equations to implement a stable nonlinear attractor system capable of generating rhythmic and discrete trajectories, which are controlled via low-dimensional parameters specifying the motion's shape and goal. Policies Modulating Trajectory Generators (PMTG) (Iscen et al., 2019) learn a policy to control a predefined trajectory generator via low-dimensional parameters and also generate a residual term to be added to the trajectory generator's output. For example, to produce legged locomotion, PMTG uses equations of motion composed from a combination of sinusoidal functions and hand-engineered gait patterns, which are parameterized by stride length, walking height, and frequency. Generally, trajectory-centric methods can work well when the equations of motion capture good solutions for the desired movement; however, such equations can be difficult to design and make robust.

**Behavioral Imitation Priors**   Desired movement is encoded through learnable functions, e.g. ANNs, trained to imitate reference motions, e.g. from motion capture or manual keyframing. Neural Probabilistic Motor Primitives (NPMP) (Merel et al., 2019c) train expert policies to imitate human motion capture trajectories, then compress these experts into a single generalist policy featuring a latent-variable bottleneck, thereby creating a common embedding space that a higher-level controller can use to interpolate and combine various motor primitives. Bin Peng et al. (2020) train expert policies to imitate animal motion capture trajectories using reinforcement learning and further deploy the learned policies on a physical legged robot. Generally, imitation methods bypass the time and manual effort involved in designing equations of motion and have achieved diverse behaviors. This comes at the cost of compiling of reference motions from humans and animals. Further, learning individual expert policies does not make use of the shared structure between movements to learn more efficiently. In animal legged locomotion, for instance, neural circuit studies have suggested that different gait patterns (e.g. walk, trot, bound, gallop) could actually be different emergent dynamic modes of the same pattern generator network (Grillner & El Manira, 2020). In this case, a single policy with the right structure should be able to capture these diverse reference motions.

**Architectural Priors**   Desired movement is encoded indirectly through the structure of ANNs, establishing inductive biases to guide learning. Heess et al. (2016) decompose a policy into a low-level "spinal" network with access to only proprioceptive signals (e.g. joint positions) and a high-level "cortical" network with access to exteroceptive signals (e.g. distance to target, vision); this hierarchical architecture was loosely inspired by animal nervous systems. After pre-training the spinal network with a shaped reward, the cortical network was able to learn complex locomotion tasks from sparse rewards by controlling the spinal network, while flat baseline architectures failed. This architecture was later shown to generalize to locomotion with challenging terrains and obstacles, providing greatly improved learning efficiency (Heess et al., 2017). Generally, architectural priors can provide improved efficiency and abstraction while maintaining flexibility. However, the level of flexibility needs to be chosen appropriately. Too much flexibility can lead to learned behaviors that are not naturalistic and infeasible/dangerous to deploy in the real world without regularization (Bin Peng et al., 2020), while too much constraint can impede the learning of diverse movements.

In this work, we design an architectural prior inspired by neural circuitry. In principle, this kind of prior should scale to diverse movements, since the architecture models the exact mechanisms that animals use to achieve robust, flexible behavior. Moreover, since working "blueprints" already exist in nature, translating architectures from biology could prove more desirable and more scalable than hand-designing trajectory priors from scratch. Further, while we use reward-based algorithms to learn parameters (i.e. RL and ES), our architecture is largely agnostic to the learning algorithm, so it is possible that our architecture could also be trained within a behavioral imitation setting.

## 3   MODEL

We translate nematode (*C. elegans*) circuits for locomotion into an ANN model applied to a simulated Swimmer agent. In Section 3.1, we provide an overview of the nematode body structure and the modular neural circuits underlying locomotion. In Section 3.2, we describe our abstract integrator and oscillator units that serve as building blocks for our architecture. In Section 3.3, we formalize the observation and action space of the Swimmer agent, and we propose our network architecture modeled on nematode circuits.

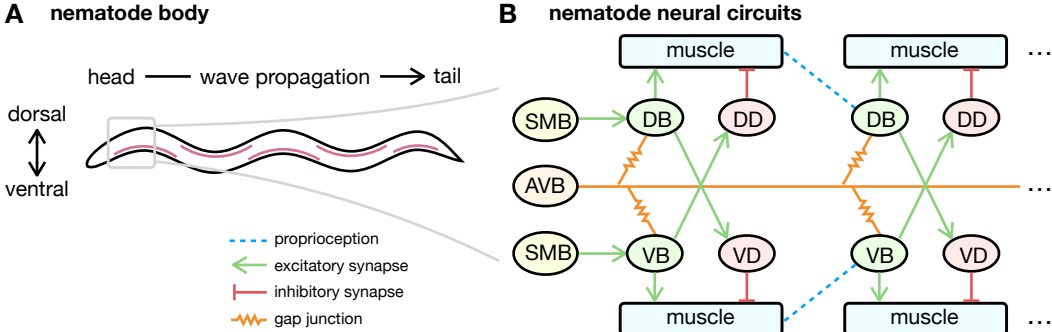

Figure 1: **Nematode. (A)** The nematode *C. elegans* achieves forward locomotion through alternating dorsal-ventral muscle contraction waves propagating down the body. **(B)** Muscle wave propagation, oscillation, steering, and speed control are coordinated by a highly stereotyped, modular, and repeated microcircuit. B neurons sense bending in the previous module via proprioception and excite ipsilateral muscles, while inhibiting contralateral muscles via D neurons. Intrinsic oscillations in B neurons initiate waves. SMB neurons bias head and neck muscles for steering. AVB attenuates all B neurons via gap junctions for speed control.

## 3.1 NEMATODE

The nematode *C. elegans* has served as a useful model organism within neuroscience because it is one of the simplest organisms with a nervous system. Moreover, it is unique in that its connectome, i.e. wiring diagram, has been completely mapped (Hall & Altun, 2008).

**Nematode Body**    The nematode body is a 1 mm long, 50 µm diameter tapered cylinder (Figure 1A). It is made up of 959 somatic cells, of which 302 are neurons comprising the nervous system, of which 75 are motor neurons that innervate the 95 body wall muscles distributed along the body. Forward and backward thrust is produced via alternating dorsal-ventral muscle contraction waves propagating down the body in the direction opposite to the direction of motion. Steering is produced by differential activation of the 20 anterior muscles in the head and neck (Gjorgjieva et al., 2014).

**Nematode Neural Circuits**    The nematode forward locomotion circuit is summarized below (Figure 1B). For a more in-depth treatment, consider Gjorgjieva et al. (2014) and Wen et al. (2018).

Muscle wave propagation is coordinated by 2 classes of neurons that innervate dorsal (D-) and ventral (V-) muscles. B neurons (DB and VB) act as both sensory and motor neurons, expressing stretch receptors in their dendrites to sense bending 200 µm anterior to their somas, and sending excitatory output (via ACh) to the muscles and to D neurons. D neurons (DD and VD) send inhibitory output (via GABA) to the muscles. This microcircuit is highly stereotyped, modular, and repeated down the length of the body, and its logic is interpretable. For a particular module, body bending in the previous module is sensed by B neurons, which then initiate bending on the same side (ipsilateral) while simultaneously inhibiting bending on the opposite side (contralateral) through D neurons.

Muscle wave initiation is generated by intrinsic oscillators. While proprioception-only circuits (with oscillators ablated) are capable of producing small waves on its own, oscillators are used initiate and entrain larger waves (Gjorgjieva et al., 2014). These oscillators were long believed to only reside in the head and neck, but recently work has shown them to in fact also be present in the body, as it is the B neurons themselves that produce intrinsic oscillations (Wen et al., 2018).

Steering is generated by the differential activation of SMB neurons biasing the head and neck muscles to bend dorsally or ventrally (Izquierdo & Beer, 2015).

Speed control is coordinated by the AVB command neuron, which is connected through gap junctions with all B neurons. When AVB is in a low state, the resting membrane potentials of B neurons are hyperpolarized to prevent activation; when AVB is in a high state, B neurons are free to activate based on proprioceptive and oscillatory inputs.

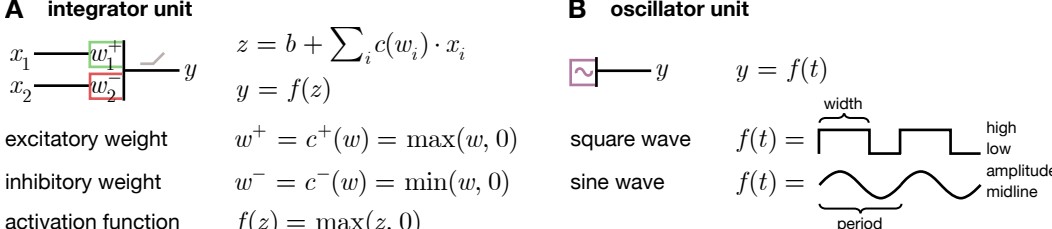

Figure 2: **Architectural Components. (A)** An integrator unit models a simple neuron. The graded input signals are multiplied by weights that represent synaptic efficacy and which are constrained to be either excitatory (positive, green boxes) or inhibitory (negative, red boxes). The graded output signal is produced from an activation function. **(B)** An oscillator unit produces driving signals much like intrinsic pacemaker cells and network-based oscillators. The graded output signal is generated through periodic functions, e.g. square waves and sine waves.

## 3.2 ARCHITECTURAL COMPONENTS

Our architecture is built from components that combine the discrete-time ANN framework that is standard in machine learning with features from computational neuroscience like constraints on synapse sign (i.e. excitation vs. inhibition) and special cell types (i.e. intrinsic oscillator). The components are fully differentiable and therefore compatible with backpropagation-based learning algorithms, though not restricted to them.

**Integrator Units**   Signals in biological neural circuits are processed and integrated by neurons. The integrator unit[1] in Figure 2A is similar to the standard ANN model. The graded inputs $x_i$ are multiplied by synaptic weights $w_i$ to produce the membrane potential $z$, given the resting membrane potential $b$. A nonlinear activation function $f(z)$ produces the graded output $y$ based on the membrane potential. Unlike the standard ANN model, however, we constrain synaptic weights by a sign constraint function $c(w)$. This is done to reflect that in biological circuits a primary characteristic of a synapse is whether it is excitatory or inhibitory. In the standard model, synapses are initialized with random signs and are free to change during learning. We argue that principled excitation/inhibition is fundamental for interpreting and modeling the logic of neural circuits, and we show in the ablation analysis that they are critical for learning in our architecture (Section 4.5).

**Oscillator Units**   Neural circuits often feature components with specialized dynamics, with oscillators being a prominent example (Grillner & El Manira, 2020). An oscillator can be implemented through coupled activity between neurons or within a single neuron, similar to pacemaker cells in the heart (Bucher et al., 2015). Oscillators serve as internal drivers of activity, a good example of the fact that neural circuits do not exclusively react to external inputs from the environment. The oscillator unit in Figure 2B uses a periodic function $f(t)$ to produce the graded output $y$. Example periodic functions include square wave and sine wave generators.

## 3.3 SWIMMER

The Swimmer is a common body design in widely adopted continuous control benchmarks, e.g. DeepMind Control Suite (Tassa et al., 2020) and OpenAI Gym (Brockman et al., 2016). We target this standard design rather than a custom body like previous work (Sarma et al., 2018; Izquierdo & Beer, 2015) in order to demonstrate the potential of using biologically inspired network architecture on tasks that have been tackled by the AI community.

**Swimmer Body**   The Swimmer agent has an articulated body with $N$ joints connecting $N + 1$ links (Figure 3A). Its movement is entirely within the $xy$-plane. Thrust is generated by the links pushing

---

[1]Simple neurons are often approximated as a single integrator units (Torres & Varona, 2012). However, sometimes neurons have multiple sites of integration, i.e. dendritic integration across multiple compartments. We prefer "integrator unit" to "neural unit" as a complex neuron may require multiple integrator units to model.

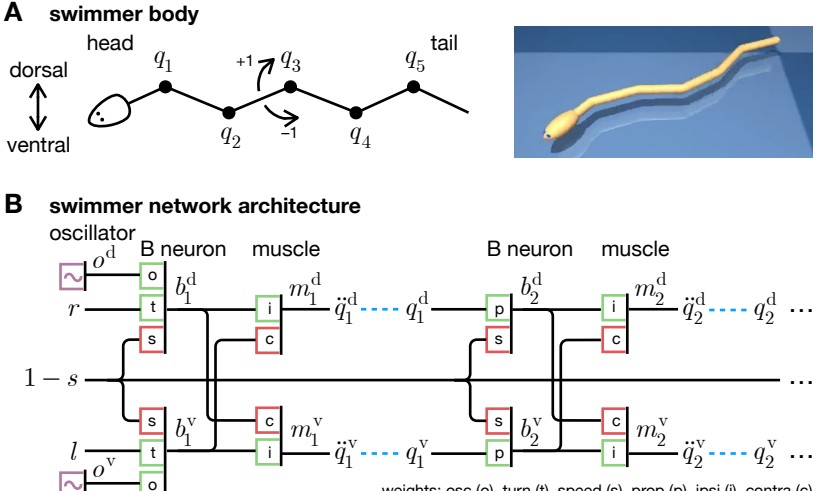

Figure 3: **Swimmer.** **(A)** The Swimmer has an articulated body with $N$ joints connecting $N+1$ links. Its observation space is normalized joint positions $q$, and its action space is normalized joint accelerations $\ddot{q}$. **(B)** Our network architecture closely conforms to the modular microcircuit of the nematode. Each module $i$ senses bending in the previous module $q_{i-1}$ and drives B neurons $b_i$ and muscles $m_i$, which are combined to create joint accelerations $\ddot{q}_i$.

against the surrounding fluid, e.g. simulated via a high-Reynolds fluid drag model (Todorov et al., 2012). The observation space consists of normalized joint positions $q \in [-1, 1]^N$ between joint limits. The action space consists of normalized joint accelerations $\ddot{q} \in [-1, 1]^N$ between maximum acceleration counterclockwise and clockwise, respectively.

**Swimmer Network Architecture**   Our network architecture is best explained visually (Figure 3B).

For muscle wave propagation, signals are integrated in B neurons and muscles; D neurons mainly serve to convert opposite-side B neuron signals from excitatory to inhibitory, and their role can be replicated directly in the muscle integrator units. We model $N$ modules to control each of the $N$ joints. For a particular module $1 \le i \le N$, the previous module joint position $q_{i-1}$ is split into dorsal $q_{i-1}^d \in [0, 1]$ and ventral $q_{i-1}^v \in [0, 1]$ components, in order to mirror signals from proprioceptive stretch receptors that are sensitive to bending on one side. B neurons are modeled as integrator units with outputs $b_i^d$ and $b_i^v$, which receive same-side excitatory proprioceptive inputs. Muscles are modeled as integrator units with outputs $m_i^d$ and $m_i^v$, which receive same-side (ipsilateral) excitatory B neuron input as well as opposite-side (contralateral) inhibitory B neuron input. Finally, the joint acceleration $\ddot{q}_i$ is calculated from dorsal and ventral muscle outputs, which act antagonistically.

For muscle wave initiation, the first module B neurons $b_1^d$ and $b_1^v$ receive inputs from oscillators $o^d$ and $o^v$, respectively, instead of proprioception. We use square wave generators acting in anti-phase.

For steering, SMB outputs are modeled as a right turn signal $r \in [0, 1]$ and a left turn signal $l \in [0, 1]$, which serve as additional excitatory inputs to first module B neurons $b_1^d$ and $b_1^v$, respectively.

For speed control, AVB outputs are modeled as a speed signal $s \in [0, 1]$. To approximate the effect of gap junctions, such that $s = 0$ represents stopping and $s = 1$ represents maximum speed, $1 - s$ serves as an additional inhibitory input to all B neurons.

The complete architecture for module $i$ is therefore:

$$q_{i-1}^d = \max(q_{i-1}, 0) \qquad\qquad q_{i-1}^v = \max(-q_{i-1}, 0)$$

$$b_i^d = f\left(w_{prop}^+ q_{i-1}^d + w_{speed}^-(1 - s)\right) \qquad b_i^v = f\left(w_{prop}^+ q_{i-1}^v + w_{speed}^-(1 - s)\right)$$

$$m_i^d = f\left(w_{ipsi}^+ b_i^d + w_{contra}^- b_i^v\right) \qquad m_i^v = f\left(w_{ipsi}^+ b_i^v + w_{contra}^- b_i^d\right)$$

$$\ddot{q}_i = m_i^d - m_i^v$$

with special B neuron integrator units for $i = 1$:

$$b_1^{\mathrm{d}} = f\left(w_{\mathrm{osc}}^{+}o^{\mathrm{d}} + w_{\mathrm{turn}}^{+}r + w_{\mathrm{speed}}^{-}(1-s)\right) \qquad b_1^{\mathrm{v}} = f\left(w_{\mathrm{osc}}^{+}o^{\mathrm{v}} + w_{\mathrm{turn}}^{+}l + w_{\mathrm{speed}}^{-}(1-s)\right)$$

We use weight sharing such that weights with the same name are shared across modules as well as within each module across sides. We initialize all weights with the correct signs and magnitudes of 1.

## 4 EXPERIMENTS

**Learning Formalization**  We consider an agent formalized as a policy function $\pi_{\boldsymbol{\theta}}(\boldsymbol{a}_t|\boldsymbol{s}_t)$ that maps states $\boldsymbol{s}_t$ to actions $\boldsymbol{a}_t$, and which is represented by an ANN parameterized by weights $\boldsymbol{\theta}$ with our architecture described in Section 3.3. We consider the standard agent-environment interaction model formalized as a Markov Decision Process (MDP). At every timestep $t$, the agent in state $\boldsymbol{s}_t$ takes an action according to its policy $\boldsymbol{a}_t \sim \pi_{\boldsymbol{\theta}}(\boldsymbol{a}_t|\boldsymbol{s}_t)$, receives a reward $r_t$, and transitions to a new state $\boldsymbol{s}_{t+1}$. Policy parameters $\boldsymbol{\theta}$ are optimized to maximize the discounted return $\sum_{t=0}^{T}\gamma^t r_t$, where $T$ is the horizon of the episode, and $0 < \gamma \leq 1$ is the discount factor.

**Learning Algorithms**  We compare backpropagation-based RL algorithms and a derivative-free ES algorithm for learning parameters in the architecture.

- **Proximal Policy Optimization** (PPO): (Schulman et al., 2017) A model-free, on-policy, policy gradient RL method. It uses a clipped surrogate objective to limit the size of policy change at each step, thereby improving stability. Since it assumes stochastic policies, we perturb the deterministic actions with Gaussian noise $\epsilon_i \sim \mathcal{N}(0, \sigma^2)$.
- **Deep Deterministic Policy Gradient** (DDPG): (Lillicrap et al., 2019) A model-free, off-policy, policy gradient RL method. It uses off-policy data and the Bellman equation to learn the Q-function, which is iteratively used to improve the policy.
- **Evolution Strategies** (ES): (Salimans et al., 2017) An evolutionary black-box optimization method. It creates a population of policy parameter variants through perturbations with Gaussian noise, then combines them through averaging, weighted by the return collected across episodes.

**Learning Setup**  We implement the Swimmer using the standard $N = 5$ body in the DeepMind Control Suite (Tassa et al., 2020) built upon the MuJoCo physics simulator (Todorov et al., 2012). We train the agent to swim using shaped rewards proportional to swimming speed (Appendix A.1).

### 4.1 PERFORMANCE AND DATA EFFICIENCY

*How well and data efficiently does learning occur in NCAP vs. MLP architectures?*

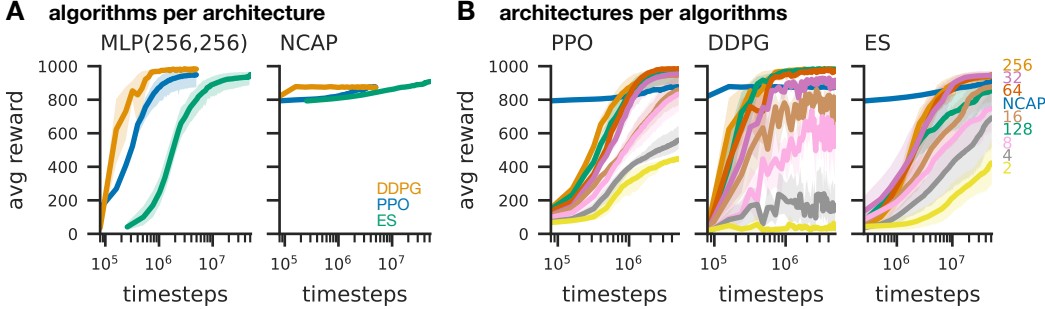

Figure 4: **Performance and Data Efficiency.** (A) Comparison of different algorithms for each architecture. Our architecture starts with high reward and improves with learning, achieving significantly better data efficiency and comparable performance. (B) Comparison of different architectures for each algorithm. Our architecture with 4 parameters overperforms small MLPs (MLP(2,2) has 70 parameters) and is comparable to large MLPs (MLP(256, 256) has 77,222 parameters). Plots show averages over 10 random seeds (solid lines) and 95% bootstrap confidence intervals (shaded areas).

First, we compare different learning algorithms for either MLP or NCAP architectures (Figure 4A). We use an MLP with 2 hidden layers of dimensions (256, 256) and ReLU nonlinearities. We find that our NCAP architecture achieves substantially higher initial performance than MLPs as well as comparable asymptotic performance with MLPs, demonstrating the effectiveness of prior knowledge encoded in network architecture. Our NCAP architecture shows reduced variance during learning between trials with different random seeds, as well as reduced differences in asymptotic performance between algorithms. For both MLP and NCAP, ES requires roughly an order of magnitude more data to achieve comparable performance with the RL algorithms, consistent with previous work (Salimans et al., 2017). Qualitatively, both MLP and NCAP architectures yield reasonable swimming movement (Videos 1A-B), though NCAP produces waves with large amplitudes resembling *C. elegans* movement, while MLP produces waves with small amplitudes more resembling tadpoles. This different movement shape explains the slightly lower asymptotic performance for NCAP because the body's direction of travel is less correlated with the head orientation, which is relevant for how rewards are calculated (Appendix A.1, Videos 1C). We note that we simplified our design from actual *C. elegans* circuits for pedagogical reasons (Section 4.5), and our goal is not to solve this swimming task *per se* but rather to investigate the advantages of biologically inspired network architecture more generally. *C. elegans* circuits are not optimized for fast swimming with few segments (Section 4.4); future work may propose architectures better for this specific task, e.g. using larval zebrafish circuits.

Second, we compare different architectures for each learning algorithm (Figure 4B). We use MLPs with 2 hidden layers of varying dimension sizes and ReLU nonlinearities. We find that performance deteriorates across all algorithms for MLPs as hidden dimensions become smaller, with some algorithms like DDPG deteriorating dramatically. However, our NCAP architecture with 4 parameters overperforms small MLPs and is comparable to large MLPs. This is especially notable as the smallest MLP(2,2) has 70 parameters (1 order of magnitude more than NCAP) and the largest MLP (256,256) has 73,222 parameters (2 orders of magnitude more than NCAP). This suggests that the relatively simple structure of our NCAP architecture provides highly effective inductive biases.

## 4.2 PARAMETER EFFICIENCY

*How valuable are parameters in NCAP vs. MLP architectures?*

We compare the asymptotic performance divided by parameter count for each architecture (Figure 5). Across all algorithms, our NCAP architecture achieves more than an order of magnitude better parameter efficiency than MLPs of any size. A parameter in our NCAP architecture is "worth more" than one in MLPs.

## 4.3 INTERPRETABILITY

*How interpretable are unit dynamics in our NCAP architecture?*

We visualize the dynamics of each network unit while performing a task (Videos 2). Since our NCAP architecture is a sparsely connected, modular network where units play constrained roles (e.g. excitatory or inhibitory), it is easier to relate unit dynamics to agent behavior, which helps in explanation and debugging.

## 4.4 TRANSFER

*How well does our NCAP architecture adapt to new bodies?*

We leverage the modular structure of our NCAP architecture to adapt a trained network for $N = 5$ joints to bodies with different $N'$ joints by duplicating/removing modules (Figure 6). We achieve robust swimming behavior for a wide range of $N'$ (Videos 3). Interestingly, zero-shot performance is *better* for longer bodies, likely reflecting that *C. elegans* circuits are evolved for its more segmented body. This kind of zero-shot transfer is not possible with monolithic, densely connected MLP architectures.

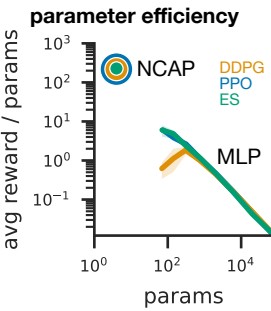

Figure 5: **Parameter Efficiency.** Asymptotic performance per unit parameter. Our architecture achieves better parameter efficiency than MLPs of any size.

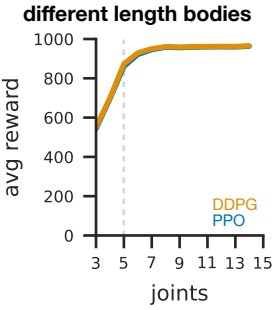

Figure 6: **Transfer.** Zero-shot transfer to new bodies after training on $N = 5$ by leveraging architecture modularity.

Figure 7: **Ablations.** **(A)** Ablations of weight sharing, sign constraints, and weight initialization in different combinations. Sign constraints (i.e. principled excitation/inhibition) are crucial for learning. Weight initialization at large magnitudes is responsible for good initial performance. Weight sharing yields a small gain in data efficiency. **(B)** Ablation of sparse connectivity yields an equivalently sized MLP (Appendix B). Learning is restored without sparse connectivity.

## 4.5 ABLATIONS

*What are the effects of various features of our NCAP architecture on performance and learning?*

First, we investigate the role of weight sharing, sign constraints, and weight initialization (Figure 7A). Without weight sharing, weights across modules and across sides are separate parameters, increasing the total number of parameters from 4 to 30. Without sign constraints, the identity function is applied instead of excitatory/inhibitory constraint function. Without weight initialization, weight magnitudes are initialized through a uniform random distribution within $[0, 1]$, rather than at 1. If using sign constraints, weights are always initialized with the appropriate sign; otherwise, signs are chosen randomly with equal probability. We find that sign constraints are crucial for learning. Without appropriate sign constraints, the NCAP architecture fails to learn during the allotted timesteps, for both RL and ES algorithms. With appropriate sign constraints, even if the weights magnitudes are not initialized ideally, the NCAP architecture will learn. Weight initialization is responsible for good initial performance. Weight sharing has a smaller, but identifiable, contribution to data efficiency.

Second, we investigate the role of sparse connectivity that arises from the natural structure of neural circuits (Figure 7B). Our Swimmer architecture has the special property that it can be completely embedded within an MLP with 3 hidden layers of dimensions (12, 10, 10) and ReLU nonlinearities (Appendix B). Specifically, after ablating sign constraints and weight sharing, our architecture is identical to this MLP with highly pruned connectivity (mostly weights of 0). We remove this sparsity and find that the MLP can learn the task with similar asymptotic performance as NCAP.

Taken together, our results suggest that principled excitation/inhibition is an important design consideration in small, sparse architectures like our NCAP, but less important in MLPs. This may be related to the "Lottery Ticket Hypothesis" (Frankle & Carbin, 2019), which suggests that, upon initialization, the MLPs already contain subnetworks with initial weights *and signs* that do most of the work for learning, i.e. they are "winning tickets"; imposing sparsity eliminates these overlapping subnetworks.

## 5 DISCUSSION

We asked what advantages biologically inspired network architecture can provide in the context of motor control. Through our case study translating *C. elegans* locomotion circuits into an ANN model, we found that biologically inspired network architecture can achieve comparable asymptotic performance to MLPs with significantly improved initial performance, data efficiency, parameter efficiency, interpretability, and transfer. Therefore, while the norm of using *tabula rasa* architectures like MLPs may be general, architectural priors can provide useful inductive biases for motor control and should be investigated further. Future work can translate neural circuits that underlie a wider variety of animal bodies and movements, as well as incorporate inductive biases from neural circuits that underlie visual, tactile, and auditory sensing. Overall, we believe that our work suggests a way of advancing artificial intelligence and robotics research inspired by systems neuroscience.

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

## A EXPERIMENTAL DETAILS

### A.1 TASKS

The `swim` task aims to test the agent's ability to swim forwards at a desired speed. It returns a smooth reward that is 0 when stopped or moving backwards, and rises linearly to and saturates at 1 when swimming at the desired speed.

### A.2 IMPLEMENTATION

**Libraries**   Neural networks were implemented in PyTorch (Paszke et al., 2019). The RL algorithms were implemented using Tonic (Pardo, 2021). The ES algorithm was implemented using ES Torch (Karakasli, 2020).

**Computational Resources**   Training was performed on a high performance computing cluster running the Linux Ubuntu operatin system. RL algorithm training runs were parallelized over 8 cores, while ES algorithm runs were parallelized over 32 cores.

### A.3 HYPERPARAMETERS

**RL Algorithms**   Standard hyperparameters for PPO and DDPG in Tonic (Pardo, 2021) at commit `48a7b72`; timesteps, 5e6.

**ES Algorithm**   Population size, 256; noise standard deviation $\sigma$, 0.02; L2 weight decay, 0.005; optimized, Adam; learning rate, 0.01; timesteps, 5e7.

**NCAP Swimmer**   Oscillator, square wave, period 60 timesteps, width 30 timesteps.

# B    SWIMMER ARCHITECTURE DETAILS

Our NCAP architecture has the special property that in can be completely embedded within a fully connected MLP of 3 hidden layers and ReLU nonlinearities. This enables us to "interpolate" between our NCAP architecture and the MLP architecture, conducting a fine-grained analysis of how various features of our architectural prior contribute to performance and learning.

By rearranging terms in the Swimmer network architecture diagram (Figure 3B), we arrive at the following network ($N = 5$ shown):

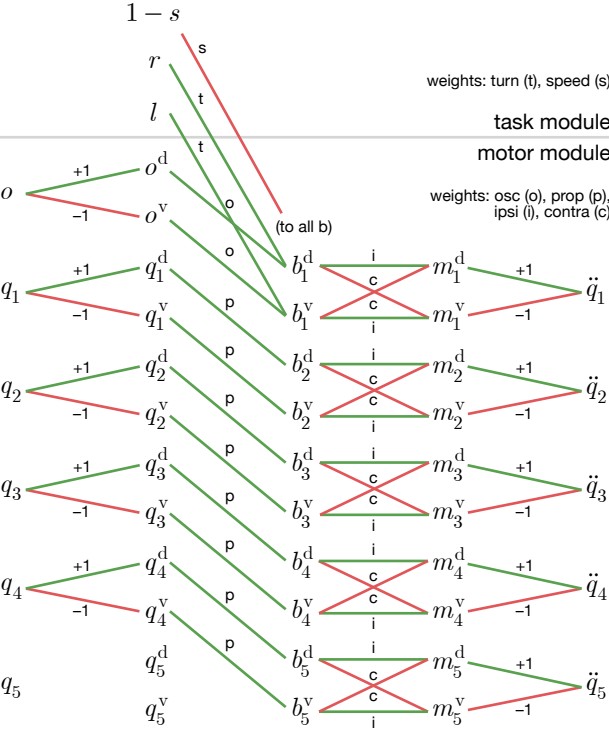

By removing weight sharing, sign constraints, and sparse connectivity, we arrive at a fully connected MLP of 3 hidden layers ($N = 2$ shown):

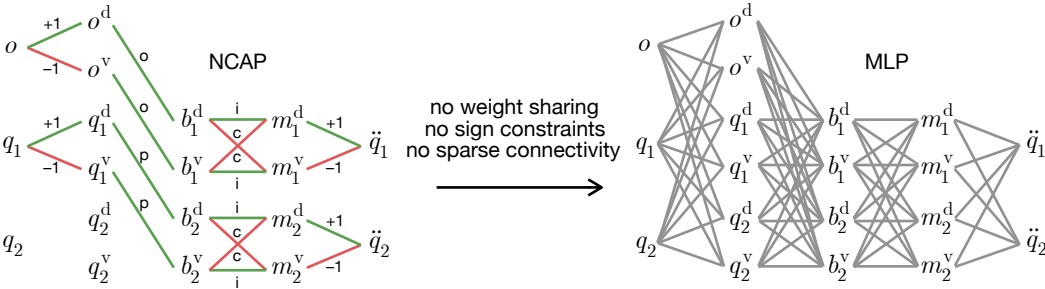

For $N = 5$, the resulting MLP has hidden layers of dimensions (12, 10, 10).

