# OpenReview forum: "Neural Circuit Architectural Priors for Embodied Control"
_ICLR.cc/2022/Conference — ICLR 2022 Submitted_

### Official Review · Reviewer_GaKc · 2021-10-30

**Correctness:** 2
**Technical Novelty And Significance:** 2
**Empirical Novelty And Significance:** 2
**Recommendation:** 3
**Confidence:** 4

**Main Review:**

### Clarity:
This paper is well written and the authors clearly explain the motivations, and framework in which they plan to operate. Figure 1, seems a bit unnecessary as there is nothing particularly new and the font size is small and exactly repeats the text. Fig. 3, and 4 are hard to read because of the font size and the organization. We would suggest the authors reorganize the figures for more clarity, which would help with the explanation of the results as well. However, Figure 2 is very clear and the parallel with C.elegans is made clear.


### Strength:
This paper asks an important question that is how to combine knowledge of biological neural circuits and artificial neural networks in a principled way. It uses an interesting example that is C.elegans for which a lot is known about it's neural architecture and principles of locomotion. It uses unconventional neural components to the machine learning community and makes it differentiable, thus possible to train with standard learning methods.

### Weakness:
Hierarchical component models for building biologically inspired neural networks are not new. In particular, the entire connectome of C.elegans has been reconstructed, and little of that information is actually replicated (in the prior) in this work.
Moreover, although the neural architecture is somewhat biologically plausible, the problem of weight sharing is not. It is also true for the learning rules used. Thus the use for understanding biological models is very limited, as it only vaguely replicated the neural structure but does not operate in a biological setting.

From the experiments, it seems that training does almost nothing to the performance of the model, as show in Fig.3A and 3B. The model initial outperforms competing models but after training it underperforms it. The relevance of the model is thus questioned, it is only the neural structure that implies the performance and not the learning. As a result this type of model would only be useful for extremely simple organisms with few neurons like C.elegans with little learning/genetic encoding ratio.









his allows us to design small, sparse networks that mirror neural circuits
yet are fully differentiable and therefore compatible with backpropagation-based learning algorithms.
In contrast to previous work on neuromechanical models of movement (Sarma et al., 2018; Boyle
et al., 2012; Izquierdo & Beer, 2015), our models are more abstract, include learned parameters, and
are used to control bodies significantly different from the originals in terms of mechanics, degrees of
freedom, and actuators.

We apply NCAP to translate C. elegans locomotion circuits into a simulated Swimmer agent, which we
train using both reinforcement learning (RL) and evolution strategies (ES). We show that our NCAP
architecture achieves comparable asymptotic performance with fully connected MLP architectures,
while dramatically improving data efficiency and requiring far fewer parameters. We further show
through an ablation analysis that principled excitation/inhibition and initialization play significant
roles in our NCAP architecture.

**Summary Of The Paper:**

This paper proposes both a set of artificial neural components and interesting principles for producing biologically-inspired neural networks for embodied control. This work aims to be at the intersection between neuroscience and machine learning for improving the design of artificial neural networks as well as improving our understanding of observed biological networks. Various components of biological networks are replicated in their framework, e.g., the balance between excitation and inhibition, intrinsic oscillators, or sparsity. The model is evaluated on C.elegans motion locomotion circuit using different types of training strategies.


**Summary Of The Review:**

This paper is interesting but does not proposes a significant improvement to the literature as the gap between the promises made in the motivation and actually delivered work is too wide.

From a neuroscience point of view, this work does not provide substantial evidence of the importance of the model at either modeling or simulating biological neural systems.

From a "theoretical" point of view, the model does not provide much advancement to the machine learning community either.

---

> ### Author Response · Authors · 2021-11-23
> **Author Response to R4**
>
> Thank you for your feedback! In addition to the general comments in our Author Response, here are some more specific responses:
>
> **Hierarchical models.** We agree that hierarchical models are not new and don’t really make that claim; rather, the core contribution of our work is an investigation of the performance, data efficiency, parameter efficiency, interpretability, and transfer potential of biologically inspired network architecture for motor control.
>
> **Biological plausibility.** Ultimately, our goal isn’t to propose a biologically plausible model of brain systems, but rather to use brain systems to inspire better ANN architectures. So we agree that DDPG/PPO isn’t biologically plausible, but don’t think this is a major issue. Our results show significant advantages of biologically inspired network architecture over MLPs and motivates more research on ANN architectures for motor control, which is a current gap.
>
> **“It is only the neural structure that implies the performance and not the learning”.** See our responses to R1 (“Hand-crafted architecture does most of the heavy lifting”) and R3 (“Weaker results compared to some baselines”). Those responses could likely address your concerns.

---

> > ### Comment · Reviewer_GaKc · 2021-11-24
> > **Response to authors**
> >
> > I thank the authors for their response and for the efforts made to improve their manuscript. However, after reading the answer to my concerns and from the other reviewers, I am still convinced that this paper doesn't propose enough novelty compared to existing pieces of work. I am also not convinced by the results presented for supporting the claims made in the paper.
> >
> > The authors mention that biological plausibility is not a concern, which I understand. Yet bio-inspiration is, and they try to make a case based mainly on the simplest of all organisms for which a reconstructed connectome exists. Another reviewer also pointed the generalizability, but I would say the meaning of the results is heavily based on "hypothetical," but certainly made in good faith.
> >
> > I maintain my score.

---

### Official Review · Reviewer_qSdv · 2021-10-30

**Correctness:** 3
**Technical Novelty And Significance:** 2
**Empirical Novelty And Significance:** 2
**Recommendation:** 3
**Confidence:** 4

**Main Review:**

Pros

1. The paper is well written, especially the abstract, introduction, and related work sections. The work is correctly motivated since efficient learning is observed in the brain, and inspiration from biology is essential to improve existing learning-based methods.

2. The paper proposes a bio-inspired network trained by DRL methods while using much fewer parameters than an MLP. This shows that architectural priors induced by knowledge from biology can replace the pure-learning approach to some degree when the prior is correctly constructed.

3. The ablation studies performed by the authors demonstrated the effectiveness of the learning on the proposed network for the swimmer agent. It is interesting that the network can still work after learning with random init weights.

Cons

1. The set of NCAP and the network proposed by the paper share many similarities with existing CPG works. It is better to present the proposed approach as a fine-tuning method for CPG networks than as a general solution for robotics. Biological locomotor CPGs have inspired the design of many artificial networks for robot locomotion (for example, [Matsubara, Takamitsu, et al. 2006], [Lele, Ashwin Sanjay, et al. 2020], [Yao, Chenpeng, et al. 2021], [Lopez-Osorio, Pablo, et al. 2021]). These CPG works are based on similar concepts as the set of NCAPs proposed in this paper and explore different ways to fine-tune the CPG network. Though the fully differentiable network proposed in this work is interesting, its novelty and contribution are limited compared to previous works in the field.

2. Since the proposed approach is similar to CPG-based works, they share the same problems as well. First, these approaches are more suitable for producing stable rhythmic patterns in controlled environments, limiting their use in general real-world robot tasks. Second, as already listed in the discussion of the paper, the network's design heavily relies on domain knowledge and cannot be generalized across different tasks.

3. The authors performed experiments only on a 6-link swimmer in a simulated environment. This fails to show: 1) How can the NCAP approach be generalized to other robot tasks with different moving patterns? 2) How can the trained model adapt to more realistic environments compared with MLP-based approaches? The authors need to introduce additional case studies and experiments to prove that the proposed method is general enough to be applied to a broad spectrum of robot tasks.

4. The NCAP approach introduces strong heuristics to the network design, limiting the network's maximum performance. Thus, compared with MLP-based approaches, the NCAP-based network has comparable but poorer performance. The authors may need to think about relaxing the heuristics used in the network to improve the performance and generalizability. For example, the second ablation done by the authors in the ablation studies section might be a good direction by relaxing the excitation/inhibition restriction.

5. Other than fewer parameters, the paper failed to explore other advantages of the NCAP-based network compared with the MLP. One potential advantage can be the generalizability of the network with shared weights on different links of the swimmer. The authors can perform experiments on swimmers with different numbers of links using the same trained weight from the 6-link swimmer. The MLP cannot achieve this since the fully-connected network needs to be trained from scratch whenever input/output dimension changes.

Minor Problems

1. Equation f(x)=max(w, 0) in Fig. 2A should be f(x)=max(z, 0).

2. Fig. 3D is not clear. Different connections and neural units need more detailed legends and captions.

Matsubara, Takamitsu, et al. "Learning CPG-based biped locomotion with a policy gradient method." 2006.

Lele, Ashwin Sanjay, et al. "Learning to Walk: Bio-Mimetic Hexapod Locomotion via Reinforcement-Based Spiking Central Pattern Generation.", 2020.

Yao, Chenpeng, et al. "Humanoid adaptive locomotion control through a bioinspired CPG-based controller." 2021.

Lopez-Osorio, Pablo, et al. "Neuromorphic adaptive spiking CPG towards bio-inspired locomotion of legged robots." 2021.




**Summary Of The Paper:**

This paper proposes a set of bio-inspired Neural Circuit Architectural Priors (NCAP) as ingredients for building trainable networks for continuous control. The authors gave one example of translating the C. elegans locomotion circuits to a network using NCAP to control a swimmer agent in a simulated environment. The proposed network trained by deep reinforcement learning methods (DRL) and evolution strategies (ES) showed comparable performance and more efficient computation with fewer parameters.

**Summary Of The Review:**

Overall, the reviewer recommends not accepting the paper. Data efficiency and generalizability are interesting and critical problems for deep learning approaches to robotics. Although the work is correctly motivated, the authors failed to prove that the proposed method is general enough to be applied to a broad spectrum of robot tasks. In addition, the proposed approach shares a lot of similarities with works on central pattern generator (CPG), due to which NCAP has very limited novelty. More detailed issues with the paper are listed in the cons in main review.

---

> ### Author Response · Authors · 2021-11-23
> **Author Response to R3**
>
> Thank you for your feedback! In addition to the general comments in our Author Response, here are some more specific responses:
>
> **1. Novelty wrt previous CPG work.** Our new framing may allay some of these concerns, but we would also like to add that we don’t really consider our work to be an instance of the CPG literature you bring up (which was also quite diverse and covered spiking neural networks, continuous time networks, biped/quadruped locomotion, etc.). Rather, we think our work is most related to ANN architectural priors, like convnets. Yes, we borrow some insights from CPG work, but the primary novelty of our approach is that we work largely within the discrete-time ANN framework prevalent in AI research (that ICLR would be most familiar with) while borrowing ideas from computational neuroscience. The property of our model being a differentiable ANN is actually core to our ability to directly compare with MLPs, as well as well as “interpolate” between our architecture and MLPs (Appendix B).
>
> **2. CPG limitations.** We feel that these arguments might be a bit broad. Some “CPG-based” work, like Policies Modulating Trajectory Generators (PMTG) [Iscen et al, 2019], is to our knowledge a state-of-the-art method for quadruped locomotion that has validated its robustness both in simulation and on real-world robots. Further, trajectory priors are also difficult to generalize across “tasks” (i.e. bodies?), but that hasn't stopped their widespread use. Priors often have the effect of constraining generalization at the benefit of improving performance and efficiency. (Though we show some interesting Transfer results where our prior actually improves generalization to new bodies, see #6!).
>
> **3. NCAP generality.** Our new framing (see Author Response) should address this concern.
>
> **4. Weaker asymptotic performance.** See our response to R2 (“Weaker results compared to some baselines”). We think you’re right about the inductive biases of the C. elegans inspired architecture to being ideal for fast swimming, but we don’t think this is a big deal for our research question and contribution.
>
> **5. Other advantages.** Per your suggestion, we expand the advantages analysis to include Transfer to longer bodies, as well as Interpretability. Thanks! Also, you might find our Transfer results to be particularly surprising, as performance actually *improves* for more segmented bodies. See Section 4 our analysis of why this might be the case.
>
> **6. Minor comments.** Figures have been redesigned and equations corrected!
>
> We appreciated your detailed comments and suggestions, and hope that this revised draft will better communicate the novelty and significance of our work.

---

### Official Review · Reviewer_AX4S · 2021-10-30

**Correctness:** 2
**Technical Novelty And Significance:** 2
**Empirical Novelty And Significance:** 1
**Recommendation:** 3
**Confidence:** 4

**Main Review:**

The paper investigates a method to introduce architectural priors for neural networks for control. These architectural and design choices need to be strictly adapted to the considered agent. For example, the authors accurately describe how their method has to be used for a Swimmer agent. I appreciate how the authors accurately explain the meaning of each component of their method. I also like the choice of the Swimmer agent as a case study, because it is clear how the method can be used for this use case, and the explanation provided by the authors is clear and informative.

Despite the good quality of writing and the clarity of the explanation, I have serious doubts about the contribution provided by this paper. As also stated by the authors, the proposed method has the limit of being strongly dependent on the knowledge of neuroscience and biology; however, most RL and AI researchers are not experts in these fields. I consider this a major drawback of this method, because it strongly limits the impact of this work. I do not think I would be able to use the proposed method in another scenario. Another drawback of the paper is that the claims about sample-efficiency and sparsity are not supported by any theoretical guarantee. It is fairly intuitive that these properties may hold empirically, but stronger motivations would be preferrable, especially because I think they strongly depend on the correctness of the architectural and design choices.

The experiments are fair results, far from impressive. I understand the importance of the evidence of sample-efficiency due to the big jump start, but I am concerned about the weaker performance compared to some baselines. How can the proposed method, that heavily designs a neural network to fit the considered agent, be outperformed by a “general-purpose” network? I understand that the outperforming network is significantly bigger, but it is not impractically big. In fact, it is a reasonable size that allow efficient forward-backward propagation on regular GPUs.

Overall, for the mentioned reasons, I find this paper to be significantly lower than the acceptance threshold.

**Summary Of The Paper:**

This paper introduces a set of architectural components and design principles for constructing neural networks for embodied control. The proposed method is biologically inspired, and can be used to build neural networks specifically designed for the considered agent, e.g., the swimmer agent studied in the paper. The paper provides a literature review, describes the method, and evaluates it in an ablation study using the DDPG algorithm.

**Summary Of The Review:**

I think this paper has significant drawbacks that outweigh its strengths. The proposed method is very difficult to use, and although the authors do a good job explaining the single use case, the paper would greatly benefit from the study of more use cases. Despite the good quality of the writing and the clarity of the explanations, I think this paper should be rejected for the lack of usability of the proposed method, the absence of a theoretical study, and the need for a larger number of use cases.

---

> ### Author Response · Authors · 2021-11-23
> **Author Response to R2**
>
> Thank you for your feedback! In addition to the general comments in our Author Response, here are some more specific responses:
>
> **Swimmer case study.** We are glad that you liked the Swimmer agent as a case study and found the explanation to be clear and informative. For our reframed paper, we focus squarely on this swimmer to answer our research question, and we have provided additional advantages that you might find interesting. We hope that this case study will serve as a first step to raise awareness of the potential of biologically inspired ANN architecture for motor control.
>
> **“Strongly dependent on the knowledge of neuroscience and biology.”** While we agree that most RL/AI research are not experts in neuro, we don’t think that this is a “major drawback” that merits rejection. For one, this is a rather general criticism that could be levied at much of the subfield of Neuro-AI [Hassabis et al. 2017]. We also likely wouldn’t reject a method because it builds on technical insights (like difficult mathematics) that many RL/AI researchers are unfamiliar with. We further hope that our clear explanation in the case study can show the uninitiated reader that these concepts are not completely out of reach, and we think that an intro systems neuroscience course should suffice for understanding our approach (expertise not required). If anything, perhaps the unfamiliarity of mainstream RL/AI researchers with neuro ideas is all the more reason why this paper would be novel and insightful for the ICLR community?
>
> **“Claims about sample-efficiency and sparsity are not supported by any theoretical guarantee.”** We’re not exactly sure what such a guarantee would look like, or if you still want this given our revised structure and research question. We’re not aware of it being standard for ANN architecture papers (as opposed to algorithm papers) to be able to provide such theoretical guarantees. If you think our paper would be incomplete without it, we’d be curious if you could point at some specific examples in the ANN literature with the kind of analysis that you’re looking for.
>
> **“Weaker results compared to some baselines. / How can the proposed method, that heavily designs a neural network to fit the considered agent, be outperformed by a ‘general-purpose’ network?”** We don’t design the neural network to fit the agent body; part our work’s novelty is that we tackle standard bodies in widely use continuous control AI benchmarks (in this case DM Control Suite), unlike previous approaches with custom-designed bodies. C. elegans circuits were not designed for fast swimming with this kind of body consisting of few segments. It’s quite surprising that the network transfers at all, which is why we think our results are exciting. You might find our new Transfer results especially interesting: our architecture achieves zero-shot transfer to bodies with different number of links. In fact, the architecture performed *better* when transferred to longer swimmer bodies than the original N=5 body that it was trained on. This may seem counterintuitive, but we believe this is because more segmented bodies are closer to the C. elegans body (with 95 body wall muscles) that this circuit was designed for. This result supports our arguments (in Section 4.2) that the slightly weaker asymptotic performance is not a significant problem, particularly as our research aim is not to solve swimming *per se*. MLPs might be “general-purpose”, but they don’t achieve the initial performance, data efficiency, parameter efficiency, interpretability, and zero-shot transfer that we show are advantages of our biologically inspired architecture.
>
> Overall, we hope that our reframed paper and these comments together are convincing demonstrations of the novelty and significance of our approach.
>
> References:
> Hassabis et al., “Neuroscience-inspired artificial intelligence”, 2017.

---

### Official Review · Reviewer_6cvx · 2021-10-31

**Correctness:** 3
**Technical Novelty And Significance:** 3
**Empirical Novelty And Significance:** 3
**Recommendation:** 5
**Confidence:** 3

**Main Review:**

Unfortunately they only consider this one swimmer task, and the architecture was apparently designed with this task in mind. Furthermore this is not a very challenging control tas
It seems to me that their hand-crafted architecture (figure 3D) does most of the heavy lifting to solve this task, as can be seen by figure 5A, where the blue and the brown line do not require training at all to have a decent performance.
One interesting aspect of biological circuits is that synaptic weights cannot flip sign during training. But it is not clear whether their neurons are excitatory or inhibitory, like in biological circuits. This requires in addition that all weights from a neuron to other neurons have the same sign. Figure 5A shows that only the sign constraint really matters (a lot actually, but perhaps only because they don't allow negative weight initializations without this constraint) undermining their design principles from section 3.2.




**Summary Of The Paper:**

The paper addresses a really important challenge: To understand innate contributions to neural circuits for motor control.
Its goal is to present a set of reusable architectural components and design principles for embodied control.
They show that a resulting model can learn to swim more efficiently and requires fewer parameters while achieving similar accuracy as an MLP.


**Summary Of The Review:**

I do like the direction of research, as it tries to take inspiration from biology to improve ML
However, the way I see it, what they really have is a nice model for a really specific task but not a lot beyond that.
The paper did not convince me that their design principles would be beneficial for other tasks,
It is also not clear what exactly the principles are that make this single task work well.

---

> ### Author Response · Authors · 2021-11-23
> **Author Response to R1**
>
> Thank you for your feedback! In addition to the general comments in our Author Response, here are some more specific responses:
>
> **Direction of research.** Thanks for your enthusiasm for using biology to improve AI. In addition to the general promise of Neuro-AI, we believe that our work addresses a particularly important gap: how systems neuroscience can help inspire better ANN architectures. Hopefully, this work will motivate the interested reader to learning more about how animals and brains work.
>
> **Generalization to other “tasks”.** If by tasks you mean other “bodies” like humanoids, half-cheetahs, etc., unfortunately, we don’t expect that neuro-inspired architecture for controlling one body can control a significantly different one (this is the case in nature too for real neural circuits). We describe in our Author Response how is likely outside the scope of this paper to provide multiple “species” of animal architectures and not necessary to answer our reframed research question. However, perhaps you will find our new Transfer results exciting, as we show zero-shot transfer to new swimmer bodies with different numbers of links. It was quite surprising to us that the architecture would perform *better* for longer swimmer bodies than the original N=5 body, and we provide some insight as to why. Of course, MLPs can’t achieve this kind of zero-shot transfer because they’re monolithic, not modular as our architecture is.
>
> **“Hand-crafted architecture does most of the heavy lifting” / “The architecture was apparently designed with this task in mind”.** Indeed, our architecture provides highly effective inductive biases for the swimmer, but we think this is a feature not a bug. Our network is a good “architectural prior” precisely because it confers good initial performance, much like nature endows priors via highly structured circuits in animals. Moreover, consider a competing approach like Policies Modulating Trajectory Generators (PMTG) [Iscen et al, 2019], a state-of-the-art method for quadruped locomotion which provides “trajectory priors” in the form of hand-crafted equations: these equations also do most of the “heavy lifting”; however, this is precisely why people resort to using this method to improve performance and data efficiency. Relative to hand-crafted trajectory priors, which we’ve found are quite difficult to design, architectural priors are a flexible alternative approach for specifying inductive biases, and we believe that nature already provides good “blueprints” for efficient, transferable, and modular solutions in motor control.
>
> **“Not clear whether neurons are excitatory or inhibitory”.** We think you’re asking if our model obeys Dale’s Principle, in which case, yes our network does (via the sign constraints). It might be slightly confusing based on our circuit diagram where we merge the role of D neurons with muscles, but we respect the original principled excitation/inhibition of C. elegans circuits. As you point out, we show that this is crucial for learning.
>
> Overall, we think that our new reframing might address your primary concerns, and perhaps you’ll agree that new “species” of architectures aren’t needed for what we’re trying to show?

---

### Author Response · Authors · 2021-11-23
**Author Response**

Thank you to all the reviewers for your feedback. While the initial scores are obviously not what we had hoped for, we have taken your feedback into consideration and made changes that we believe significantly improve the manuscript. Here are the major differences:

**Framing of contribution.** In general, reviewers seemed to find the work interesting and well-motivated, and the Swimmer case study “clear and informative”. The main shared criticisms seem to stem from unmet expectations about the generality of the method. We see now how our initial framing could imply that NCAP was a method ready to apply “off-the-shelf” to translate neural circuits. A couple clarifications: We do not believe that the production of biologically inspired network architecture can be automated or homogenized at this stage. Designing neuro-inspired architectures for new bodies requires non-trivial engineering effort (much like designing good architectures generally, like Convnets and Transformers). While architectures for new bodies are within the realm of future work, we believe they’re outside the scope of a single conference paper.

Instead, we see this work as a first step to motivate ANN architectures inspired by systems neuroscience, specifically for motor control/robotics. It is still common practice in robotics to use generic MLP networks (see examples in Introduction), while fields like computer vision and NLP widely experiment with different architectures (including neuro-inspired convolutional neural networks). Further, the subfield of Neuro-AI has largely focused on “cognitive neuro” contributions to AI [Hassabis et al. 2017]; our work aims at the gap exploring “systems neuro” contributions to AI, which is quite different.

To this end, we reframe our contribution and reorganize the paper, while maintaining many of the core results (and adding exciting new ones via additional experiments). The Abstract, Introduction, and Discussion will make this clear, but in summary: “In this work, we ask what advantages biologically inspired network architecture can provide in the context of motor control.” … “We found that biologically inspired network architecture can achieve comparable asymptotic performance to MLPs with significantly improved initial performance, data efficiency, parameter efficiency, interpretability, and transfer.” We believe this new framing is clearer, more well-supported, and more interesting to the ICLR community.

**Additional experiments and analysis.** We conduct additional experiments on Interpretability and Transfer: see Experiments section and Supplemental Videos (video groups 2 and 3). We have revised the writing to be more explicit about the specific advantages observed in our proposed architecture.

**Figures and captions.** We redesigned the figures to be more legible and flow with the new structure, and the captions are now somewhat standalone.

Further minor changes and comments are addressed in responses to individual reviewers.

Thank you for your reconsideration! We will continue to answer any further questions you may have.

References:
Hassabis et al., “Neuroscience-inspired artificial intelligence”, 2017.

---

### Decision · Program_Chairs · 2022-01-20

**Decision:**

Reject

**Comment:**

Meta Review for Neural Circuit Architectural Priors for Embodied Control

The motivation of this work is to address an important challenge: To understand innate contributions to neural circuits for motor control. This paper proposes both a set of reusable architectural components and design principles, and also interesting principles for producing biologically-inspired neural networks for embodied control. This work aims to be at the intersection between neuroscience and machine learning for improving the design of artificial neural networks and improving our understanding of observed biological networks. In their model, various components of biological networks are replicated (such as the balance between excitation and inhibition, sparsity, and oscillation). They show that a resulting model, inspired by C.elegans, can learn to swim more efficiently (when evaluated on the Swimmer RL environment) and requires fewer parameters while achieving similar accuracy as an MLP.

Most reviewers, including myself, recognize (and appreciate) the ambition of this work, and are excited at the goal of looking at problems from the perspectives of both system neuroscience and machine learning. The motivations of this paper are clearly explained, and the paper is well written (also the diagrams are great). I'm very excited about this work, and hope to see it succeed, but in the paper's current state (even with the revisions), I don't think it addresses the reviewers' main concerns.

After discussions and examining the paper and the reviews in detail, I feel reviewer GaKc best summarizes the main issues with the work at its current state:

- This paper is interesting but does not proposes a significant improvement to the literature as the gap between the promises made in the motivation and actually delivered work is too wide.

- From a neuroscience point of view, this work does not provide substantial evidence of the importance of the model at either modeling or simulating biological neural systems.

- From a "theoretical" point of view, the model does not provide much advancement to the machine learning community either.

So while the current work (especially in the revised state) I would consider to be an outstanding workshop paper, I cannot recommend it for acceptance at ICLR 2022. An advice I would give to the authors (as someone who publishes to ML conferences, and Sys-Neuro/Bio venues) is that for these ML conferences, it might be easier to make the narrative of the work narrower, and well-defined. If the method is supposed to demonstrate significant advantages of biologically inspired network architecture over current RL methods, the results should clearly demonstrate convincing experimental results that can persuade the (non-neuro) RL community to have interest in the method. If the method does not achieve SOTA results, then try to present the method capable of something really useful that existing RL methods simply fail at (and emphasize that as a core contribution). Conversely, if the narrative is to use a bio-inspired network to emulate biological behaviors, the method must have something important to offer for the community of people working on simulating biological neural systems.

I look forward to seeing this work improved and eventually published at a journal or presented at a conference in the future, good luck!